# A Meta-Analysis of the Current State of Evidence of the Incredible Years Teacher-Classroom Management Program

**DOI:** 10.3390/children9010024

**Published:** 2021-12-30

**Authors:** Rachel Korest, John S. Carlson

**Affiliations:** 1Kennedy Krieger Institute, Baltimore, MD 21205, USA; rachelkorest@gmail.com; 2Department of Counseling, Education Psychology, and Special Education, Michigan State University, East Lansing, MI 48824, USA

**Keywords:** teacher-classroom management, Incredible Years, meta-analysis, effectiveness, efficacy, universal prevention program, evidence-based intervention

## Abstract

This meta-analysis evaluated the current state of evidence and identified potential treatment moderators of the Incredible Years Teacher Classroom Management (IYTCM) program used to reduce externalizing and internalizing behaviors in school-aged children. Inclusion criteria involved published studies between 1984–2018 and examining the effects of IYTCM as a standalone program on teacher and/or child behavioral outcomes. We identified and narratively summarized potential moderators, which included the severity of child behavior, dosage, study design, and reporting methods. Overall, effect sizes revealed IYTCM had moderate positive effects on teachers and small positive effects on children. Narrative summaries indicated larger effect sizes in higher dosage studies and higher risk children. The results align with previous systematic reviews on the Incredible Years Parent Training (IYPT) program but this is the first study to look at the teacher training program. Overall, IYTCM seems to be an effective intervention; however, what components of this program work best, for whom, and under what conditions require further empirical investigation.

## 1. Introduction

Evidence-based classroom management strategies are essential for enhancing student social, emotional, behavioral, and academic growth [1]. To effectively implement classroom management strategies, teachers need adequate training and support from schools. However, teachers report low confidence in managing classroom disruptions, citing inadequate training [2]. With a limited emphasis on teacher classroom management (TCM) training in preservice education programs, teachers may rely on ineffective teaching strategies. This may result in poor academic outcomes and increased challenges related to students’ social-emotional development. Additionally, this can exacerbate behavioral problems in the classroom and lead to elevated levels of teacher stress and issues with teacher retention [3].

One solution for improving teacher classroom management techniques is to implement an evidence-based intervention (EBI) [4]. Educational policy initiatives, such as the Every Student Succeeds Act of 2015 [5] support the implementation of EBIs demonstrating efficacy to improve student social-emotional skills and reduce behavioral problems in the classroom [1]. Despite research supporting the use of EBIs in the school context, few schools successfully administer EBIs because of challenges in the implementation process [6]. Even though teachers are often the main implementers of EBIs, they report a lack of training, skills, and coaching necessary to implement these interventions [7].

Understanding the influence of caregivers (e.g., teachers, parents/guardians) on the development of child psychopathology is essential to the foundations of TCM. Coercion theory highlights how negative caregiver-child interactions experienced in early childhood can lead to a pattern of coercive interactions with adults outside the home and the development of behavioral problems [8]. To disrupt this learned pattern of negative interactions in children, caregivers can be trained to not only change their professional vision of classroom management [9] but also to reshape their own behaviors and emotional reactions while managing the classroom [10]. These types of shifts in caregivers’ beliefs and behaviors are essential to increase positive interactions with their children as well as improve their overall teaching effectiveness [11]. Research supports behavioral parent training programs designed to prevent and treat undesired behavior in preschool and younger elementary students. For instance, a meta-analysis of 26 behavioral parent training programs (e.g., Parent-Child Interaction Therapy) found large effect sizes for child outcomes post-treatment (ES = 0.86) and moderate effect sizes in positive parental behavior (ES = 0.44), based on parent, observer, and teacher reports [12]. Additionally, a meta-analysis of 69 behavioral training programs (e.g., Incredible Years Parent Training, Triple-P Positive Parenting) found small to moderate effect sizes (ES) for parent (ES = 0.45) and child behavior (ES = 0.42) and small follow-up treatment effects compared to a control group for both parent (ES = 0.25) and child behavior (ES = 0.21) [13].

Although several EBIs specifically focus on training parents to improve child development, fewer EBIs focus on training teachers. Since classroom-based problems may not be readily managed through parent skill training, increasing teacher training skills using an EBI could improve success for students across home and school [14]. Thus, finding an EBI that includes both a parent training and teacher training component is essential to promoting success for children presenting with or at-risk of future challenging behaviors.

### 1.1. The Incredible Years

The Incredible Years (IY) series is comprised of three interlocking empirically-supported programs targeting children, parents, and teachers [15]. The aim of this series is to prevent, reduce, and treat behavioral problems and promote social-emotional, behavioral, and academic success for young children, aged 3–8 years. This program is internationally recognized and has been researched and implemented in countries around the world [4].

For the caregiver programs (i.e., parents and teachers), two trained facilitators lead caregivers in group discussions and activities. Within this setting, caregivers have opportunities to role-play, receive feedback from facilitators, self-reflect, and view video scenes of caregivers working with children. Participants also engage in group discussions on ways to problem-solve, develop ideas for reinforcing children’s behaviors, and have opportunities to practice their skills at home or school through handouts and activities [16].

Recently, the programs within the IY series have been studied as standalone programs, however, most of the research focuses on the IY parent training program (IYPT) [17]. Although there is extensive research demonstrating the efficacy and effectiveness of the IYPT program, the efficacy of the Incredible Years Teacher Classroom Management (IYTCM) standalone program remains unclear. Vagueness surrounding the efficacy of IYTCM may be a result of the limited research available in the past; however, in the last 10 years, there have been over 16 studies conducted on IYTCM. With more research available, a meta-analysis is an ideal tool [18] to examine the current state of evidence of IYTCM. Conducting a meta-analytic review may help determine if IYTCM improves TCM skills and child behavior and may help uncover which program components are most effective for improving these outcomes.

### 1.2. Research on IYTCM

Schools all over the world have found promising benefits of IYTCM as a standalone program. Specifically, IYTCM has demonstrated positive teacher and student outcomes in the United States [19,20,21] in Wales, United Kingdom [22,23], New Zealand [24], Ireland [25], Norway [26,27,28], Jamaica [29] and Portugal [30]. Furthermore, research has shown improved teacher-classroom management strategies for low-income, Head Start, and majority African-American populations in Chicago [31] and North Carolina [32]. Ten randomized controlled trials (RCTs) with independent researchers have all provided support for IYTCM [14,23,25,29,31,33,34,35,36,37]. Studies support its use as a selective prevention program for students with severe behavioral problems [29] and a universal prevention program for students with and without behavioral problems in the classroom in general [23]. IYTCM increases teacher and childcare-worker skills, teacher confidence in classroom management, as well as improving relationships with students and parents [22,28,31,38]. Furthermore, IYTCM is efficacious in increasing prosocial and friendship skills in students identified with behavioral problems [29], and reducing externalizing problems such as aggression and hyperactivity [23].

### 1.3. Variability within the Research

Variables identified in the literature that moderate treatment outcomes include the severity of a child’s behavior, reporting methods, study design, and training. Children with clinically significant behavioral problems have larger effects than children without [13]. Previous research on IYPT found direct observations to result in stronger effect sizes than teacher-reported data, smaller effects in RCTs when compared to quasi-experimental designs (QED) and non-random assignments, and the number of intervention sessions attended to be a significant predictor of treatment outcomes [17]. Applying similar meta-analytic techniques to examine the effect sizes of these characteristics may help to determine what specific populations, methodology, and intervention characteristics in IYTCM are most beneficial [39] and possibly increase implementation success in the school context [40].

### 1.4. Aims of the Current Study

There were three aims of the current study. First, we examined the current state of evidence for the use of IYTCM techniques in improving teacher outcomes. For teacher outcomes, we examined the use of positive (i.e., praise, positive interactions, proactive discipline) and negative TCM skills (i.e., harsh, criticism) as measured by observations. Teacher buy-in also serves as a major barrier to treatment implementation. Thus, teachers’ perceived usefulness of positive TCM strategies learned from IYTCM was also measured. A second aim examined the current state of evidence for the use of IYTCM techniques in improving child outcomes. For children, we examined both externalizing behavior (i.e., disruptive behavior, conduct problems, non-compliance with teacher directions) and prosocial skills (i.e., compliance of teacher directions, cooperating and sharing with peers, friendship skills), as seen through observations and teacher-reported behavior. The third aim of the current study was to examine the variability of study outcomes within the IYTCM literature. Specifically, we examined the target child (i.e., initial severity of child behavior), reporting method, study design, and dosage to understand if potential moderators influenced the overall effect sizes of the expected outcome variables as a result of IYTCM techniques.

The current study addressed the following research questions to address our three aims:What is the overall effect size associated with improving teacher outcomes in the classroom post-intervention?What is the overall effective size associated with improving child outcomes in the classroom post-intervention?To what extent does the effect size of IYTCM differ based on target child (i.e., severity of child behavior), reporting methods, study design, and dosage?

Based on previous meta-analytic research and systematic reviews on parent training programs [13,17], we hypothesized that (a) IYTCM would have small to moderate effects on teacher outcomes overall, (b) IYTCM would have small to moderate effects on child outcomes overall, and (c) the target child, reporting methods, study design, and dosage would not moderate the overall effect size of IYTCM.

## 2. Methods

### 2.1. Literature Search Criteria and Study Identification

A systematic and comprehensive search for studies conducted between 1984 (i.e., the first published IY research study) and 2018 occurred using the following electronic databases: ProQuest, PubMed, and Web of Science. Specific search terms included “Incredible Years AND Teacher Classroom Management Program.” We entered the search terms and Boolean operators into the search simultaneously, which resulted in a total of 107 studies (See Figure 1). We also examined the Incredible Years website and references within other IYTCM papers. When examining the Incredible Years Website, the first author screened each article in the “Teacher Training” folder and in the “Researcher (article library)” Table. This resulted in 30 new articles.

Before applying inclusion criteria, the first author and an undergraduate student screened the titles and abstracts to remove irrelevent articles and duplicates, which resulted in 31 studies. To be eligible for this meta-analysis, the studies had to meet the following criteria (a) examined IYTCM as an intervention or prevention program for TCM training and/or for children with or without externalizing behaviors; (b) examined IYTCM as a standalone program; (c) included an RCT with a control group, (i.e., treatment as usual, placebo, waitlist, no treatment), quasi-experimental design, or pre-post design; (d) published in peer-reviewed journals or conference abstracts in English; (e) included essential information to calculate effect size (i.e., included pre and post means, standard deviation, and raw data in order to calculate and standardize effect sizes); (f) measured outcomes that fell into one of the categories of prosocial behavior, externalizing behavior, positive use of teacher strategies, negative use of teacher strategies, and teacher perceived usefulness of positive strategies. Excluded studies included dissertations, pilot studies, single-case designs, brief IYTCM programs (i.e., 4 sessions or less), combined with another intervention, self-administered IYTCM programs, or part of a larger study such as [41] which included the same sample of students in a follow-up report. Additionally, conference abstracts were identified, but those later found in peer-reviewed journals were excluded to avoid redundancy. Application of the inclusion criteria to the initial search resulted in 15 studies. Finally, the first author searched the references of the 15 studies identified and found 1 additional article that met the above criteria [23]. This resulted in a total of 16 articles (Figure 1).

The current study used the Preferred Reporting Items for Systematic Reviews and Meta-Analyses (PRISMA) guidelines [42]. The study reported in this article was not formally preregistered. Neither the data nor the materials have been made available on a permanent third-party archive; requests for the data or materials can be sent via email to the lead author.

### 2.2. Study Coding

Teacher outcome variables. Teacher outcome variables were coded into three constructs: teachers’ perceived usefulness of positive teacher strategies, teacher use of positive classroom management strategies, teacher use of negatives classroom management strategies. Two measures assessed teachers’ perceived usefulness of positive classroom management strategies: Teacher Strategies Questionnaire [15] and the Teacher Satisfaction Questionnaire [16]. If a study included either of these two measures, they were coded as teacher perceived usefulness of positive classroom management strategies. If studies included multiple subscales of teachers perceived usefulness, and a total subscale, the total subscale was selected. Three studies met this criterion.

Based on the measures in the IYTCM studies, we defined teacher use of positive strategies as giving clear expectations, redirecting misbehavior and displaying warmth, respect, and positive interaction between teachers and their students. We defined teacher use of negative strategies as disrespect, anger, hostility, and aggression towards students. In some studies, positive and negative strategies were included for both an entire class and as a measure of target students of both low and high risk [23]. In this instance, we chose positive and negative strategies towards the class as the purpose of this program was to serve as a universal prevention program of problem behavior. Seven studies met this criterion.

Child outcome variables. We coded child outcomes into two constructs: prosocial behavior and externalizing behavior. Based on the measures in the IYTCM studies, we defined prosocial skills as considerate, sharing, cooperating, helping, and friendship skills. Studies were included if they had a measure that fit this definition of prosocial behavior. Seven studies met this criterion.

We coded studies as externalizing behavior if they included an instrument that measured conduct problems (e.g., tantrum, lying, stealing, fighting, disobeying directions), or hyperactivity (e.g., fidgeting, restlessness, distractibility). If a study included multiple measures of externalizing behavior and a total behavior score, we chose the total behavior score to measure externalizing behavior. Nine studies met this criterion.

Moderators. Four moderators were coded for this study: target child, reporting method, study design, and dosage.

Target child. The target child was coded into two constructs for child outcomes: high risk and low risk. Studies were coded as high risk if the study included an assessment tool measuring clinical behavioral problems at baseline for both treatment and control groups (e.g., Strengths and Difficulties Questionnaire) and scores were above the clinical range defined by the study. If studies were below the clinical cut off score, the study was coded as low risk.

Reporting method. The reporting method was coded into two constructs for child outcomes only: observation and teacher-rated. Observation was defined as any measure observing child behavior, while teacher-rated was defined as a rating scale completed by the teacher which examined child behavior.

Study design. The study design was coded into two constructs for both teacher and child outcomes: *RCT* and *QED*. RCT studies were defined as studies that used an RCT design, including a stratified, matched, or block design, and were considered efficacy studies. QED studies were defined as studies that used a pre-post analysis without an RCT and were considered effectiveness studies.

Dosage. The dosage was coded into two constructs for both teacher and child outcomes: high dosage and low dosage. Studies were coded by calculating the number of sessions offered multiplied by the number of training hours. A study was coded as high dosage if they offered greater than or equal to 42 h (6 sessions, 7 h each) of training while a study was coded as low dosage if they offered less than 42 h of training. The cut-off point was determined based on the recommendation that a minimum of 42 h is required to receive a certificate of completion for IYTCM [15]. The number of hours was used to code a study as high or low dosage instead of number of sessions due to the varied number of sessions offered (e.g., 8 sessions, 4 h each). If a study stated it was a “full day” workshop instead of the exact hours, we assumed they delivered the training in a 7-h session, which is also a certificate requirement.

Study quality. To examine the quality of the study, we estimated overall study rigor using a 9-point scale, adapted from prior studies [13,17]. This covered six indicators of methodological strength. For the group assignment, studies earned 2 points for random assignment, 1 point if they were statistically equivalent despite using a nonrandom assignment, or 0 points if groups were not statistically equivalent or information about group assignment was missing. For source of information, studies earned 2 points if they used direct observation and an additional source of information (i.e., self-report for teacher outcomes, teacher rating scale for child outcomes), 1 point if they only used direct observation or used two rating scales (i.e., parent and teacher), or 0 points if they used only a self-report or teacher rating scale. For reliability, studies earned 2 points if they included a reliability measure with a Cronbach’s alpha greater than 0.7, or 1 point for each of the following characteristics: (1) blinding of participants or researchers, (2) inclusion of important demographic information of participants (e.g., sample size, ethnicity, age), (3), and treatment fidelity was judged as adequate.

Inter-rater reliability. An inter-rater reliability check was conducted with an undergraduate student. The first author and undergraduate student evaluated the 16 identified studies based on a detailed coding scheme. This coding scheme included 17 coding criteria developed by the first author that included study-level and sample-level characteristics. The study-level characteristics included publication type, study year, school location (i.e., urban or rural), school setting (i.e., preschool, Head Start, elementary school), study design (i.e., RCT, QED), intervention dosage, number of sessions attended by teachers (e.g., calculated by the range or the percentage), study quality, reporting method (i.e., observation, teacher report, self-report), targeted outcomes, and measurement tools. The sample-level characteristics included sample size of the teachers and students, mean age of the sample participants, grade and gender of the students, years of teaching experience, and target child (i.e., high-risk or low-risk behavior identified through cut-points). Inter-rater agreement was calculated for these 17 variables by dividing the number of agreements by the total number of agreements plus disagreements, multiplied by 100. The total inter-rater agreement was therefore 94%.

### 2.3. Data Extraction

Raw data extraction from the articles was obtained by hand using predefined data fields and study quality indicators. To ensure reliability and validity, an undergraduate student independently extracted data from selected studies. Inter-rater reliability was calculated by dividing the number of agreements by the total number of agreements plus disagreements, multiplied by 100. The total inter-rater reliability was 98%.

Data analysis. Throughout this process, several studies contained more than one measure of each category. Thus, multiple measures were averaged to create one effect size as multiple measures can create an issue of dependence and no measures were clearly superior to another [18].

Since there were several measures within dependent variables (e.g., observation and teacher report data for prosocial behavior), individual effect sizes were calculated when scores were reported for multiple child outcomes (i.e., observation, teacher report data) and teacher outcomes (i.e., observation, self-report data) for later data analysis. Additionally, individual effect sizes were computed for child outcome constructs (i.e., prosocial behavior, externalizing behavior) and for teacher outcome constructs (i.e., teacher use of negatives strategies, teacher use of positives strategies). Analyses were separately run for teacher and child outcomes.

As the studies contained different methods to measure the outcome variables (i.e., rating scales, observations) and several studies included a comparison of control groups with experimental groups, the effect size of each study was calculated using the standardized mean difference formula [SMD], Cohen’s *d* [18]. Additionally, because most of our articles included both pre- and post- data, we chose an alternative effect size formula, Pre-Post with Control (PPWC; [43]) to compute effect size. Using the PPWC equation improves accuracy of the effect size as it controls for any baseline differences [44].

Bias correction. Excluding unpublished studies may create an upward bias effect as studies may not be published because of a non-statistically significant *p*-value [39]. To address this issue, we intended to create a funnel plot to examine the distribution of studies and identify publication bias within the research; however, because there were less than 10 studies in each outcome variable, this was not possible [45].

The SMD formula often has an upward bias with sample sizes less than 20 [18]. Since several of the studies included in this meta-analysis have smaller sample sizes, Hedges’ *g* correction for small sample bias was applied.

Meta-analysis calculations. The data for this study were analyzed using the R mvmeta package for multivariate regression techniques as a system of analysis [46]. For teacher outcomes, positive and negative teacher strategies were analyzed together to account for dependence. Since covariance was not provided between positive and negative teacher outcomes, a sensitivity analysis was conducted at 0, 0.2, 0.5, 0.7 and 0.95 to see if this influenced the grand mean of the effect size. For child outcomes, both prosocial and externalizing measures were analyzed together to account for dependence.

A random effects model, which estimates the distribution of true effects [39], was selected to assess teacher and child outcomes for this meta-analysis. This type of model is justified when true effect size varies from study to study. Before calculating the effect size, a test of heterogeneity was conducted to determine if the effect sizes were obtained from the same population. The heterogeneity estimate was calculated with Cochran’s *Q* and the $I^{2}$ statistic, which describes the percentage of variation across studies due to heterogeneity [47]. If there is a large amount of heterogeneity, the sources of variability should be explored through a moderator analysis [18]. Thus, the secondary aim of this meta-analysis was to explore potential moderators. However, due to the small number of studies located, these results were reported narratively [48].

## 3. Results

Sixteen studies met the criteria for this analysis. A total of 9 studies (56%) with 891 teachers and 12 studies (75%) with 9252 children were included in the sample. Most of the studies (*N* = 14; 88%) were published in journal articles and had been published recently: 2015–2019 (*N* = 8, 50%), 2010–2014 (*N* = 6, 38%), and pre-2010 (*N* = 2, 13%). Additionally, although most studies were published in Western Europe (i.e., Ireland, United Kingdom, Portugal; *N* = 7, 44%), studies were published across 6 different countries (i.e., Jamaica, United States, Portugal, United Kingdom, Ireland, New Zealand). Studies with 42 h of training or more occurred in seven (44%) of the studies and studies with less than 42 h of training occurred in nine studies (56%). The total number of sessions offered ranged from 5 to 8 sessions and teachers attended greater than 70% of sessions on average (*N* = 6, 100%).

Teacher age ranged from 21 to 69 years. Teacher experience ranged between 2 and 30 years. The mean age of children in this sample was 4.85 and ranged from 3 to 8 years (*N* = 9). A total of 7 studies (44%) were identified as low risk, 4 studies (25%) were identified as high risk, and 5 studies (31%) either did not report the risk level of their students or it was difficult to determine based on the information provided. Additionally, 4 studies (25%) were conducted in an urban school district, 3 studies (19%) were in a rural school district, 6 (37%) were in both, and 3 (19%) did not report this information. Most studies (*N* = 9, 56%) were conducted in an elementary or primary school.

Most of the studies (*N* = 9, 56%) included an RCT design compared to a QED (*N* = 7, 44%). The intervention was compared to a wait-list condition in seven studies (44%) while the intervention was compared to treatment as usual in 6 studies (37%). The overall study quality ranged from 2 to 7 points based on a 9-point scale.

### 3.1. Teacher Outcomes

Only three studies had the data available to analyze teacher’s perceived usefulness of positive strategies. Due to the small sample size and different study designs included (two single group pre-post designs [SGPP] and one PPWC), a narrative summary of the results was conducted. Results from SGPP studies ranged from *g* = 0.10 to 0.88, which indicates a small to large effect size, while the PPWC study had an effect size of *g* = 0.94, which is considered a large effect size [25].

Seven studies included an observed measure of positive teacher outcomes and six studies included a measure of negative teacher outcomes (See Table 1). The results of the sensitivity analysis showed the magnitude of correlation between outcomes did not significantly influence the effect size for either positive or negative teacher outcomes (See Table 2). The mean effect size for positive teacher outcomes across correlation values ranged from *g* = 0.70 to 0.73 (95% confidence interval width ([CI width] = [1.02–1.07], *p* < 0.01), which is considered a moderate to large effect size [49]. The mean effect size across correlations for negative teacher strategies ranged from *g* = −0.50 to −0.59 (95% CI width = [0.45–0.63], *p* < 0.001), indicating a moderate effect size.

There was one study, which yielded an effect size of *g* = 2.75, which seems to be an outlier [33]. The difference in the results with and without this outlier ranged from 0.27 to 0.28 for positive teacher outcomes and 0 to 0.13 for negative teacher outcomes. Without this outlier, positive teacher outcomes had an effect of *g* = 0.42 (*SE* = 0.12, 95% CI [0.17, 0.66], *p* < 0.001) to *g* = 0.46 (*SE* = 0.13, 95% CI [0.21, 0.71], *p* < 0.001) across correlation values, indicating a moderate effect. Negative teacher effects ranged from *g* = −0.46 (*SE* = 0.12, 95% CI [−0.69, −0.23], *p* < 0.001) to *g* = −0.50 (*SE* = 0.12, 95% CI [−0.74, −0.26], *p* < 0.001) across correlation values, indicating a moderate effect.

### 3.2. Child Outcomes

Only four studies contained data to analyze the prosocial and externalizing child outcomes of observationally measured behavior (See Table 1). Due to the small sample size and different study designs (one post group means only and three PPWC designs), the results of these studies were narratively summarized. For 3 of the studies, the prosocial effect size for child behavior ranged from *g* = 0.13 to 0.93 indicating a small to large effect size. In contrast, one study found prosocial skills decreased prosocial behavior (*g* = −0.24) [23]. Externalizing effect sizes ranged from *g* = −0.15 to −0.53, indicating a small to moderate effect size.

All nine studies identified included a teacher-rated component of prosocial behavior and seven of the nine studies included a measure of externalizing child outcomes. The results of the sensitivity analysis showed the level of correlation between outcomes did not significantly influence the effect size for either prosocial or externalizing outcomes of teacher-rated child behavior (See Table 2). The mean effect size for prosocial outcomes across correlation values ranged from *g* = 0.19 to 0.21 (95% CI width = 0.25, *p* < 0.01), which is considered a small effect size. The mean effect size for externalizing outcomes across correlations ranged from *g* = −0.21 to −0.14 (95% CI = [0.27–29], *p* < 0.05), which is considered a small effect size.

Statistically significant heterogeneity was observed in both teacher and child outcomes. Results of the *Q*-test across correlation values for teacher outcomes ranged from *Q* (11) = 51.6 to 278.45 (*p* < 0.001) and the $I^{2}$ statistic values ranged from 78% to 96%. Results of the *Q*-test across correlation values for child outcomes ranged from *Q* (14) = 93.39 to 1350.00 (*p* < 0.001) and the $I^{2}$ statistic values ranged from 85% to 99%. High heterogeneity within these studies supports applying the random effects model and exploring potential moderator variables.

### 3.3. Moderating Variables

Four potential moderators were identified a priori to explain the heterogeneity in teacher outcomes and child outcomes: target child, reporting method, study design, and dosage. Due to the small number of studies, these results were narratively reported [48]. Descriptive statistics of these results can be found in Table 1 and Table 3.

Target child. Eleven studies reported whether the children in the sample were high risk or low risk (See Table 1). High risk prosocial outcomes ranged from *g* = 0.28 to 0.92, indicating a small to large effect size (*N* = 4). For low risk prosocial outcomes, effect sizes ranged from *g* = −0.24 to 0.20 (*N* = 7) indicating a small negative effect to a small positive effect size (See Table 3). For externalizing outcomes, effect sizes ranged from *g* = −0.15 to −0.56 for high risk children indicating a small to moderate effect size (*N* = 4). Effect sizes for low risk children ranged from *g* = −0.03 to −0.29, indicating a negligible difference to a small effect size (*N* = 7).

Reporting method. Four studies included observation data and nine studies included teacher-rated child behavior data (See Table 1). Observational data for prosocial child outcomes ranged from *g* = −0.13 to 0.93 indicating a small negative effect to a large positive effect. Teacher-rated prosocial outcomes ranged from *g* = −0.03 to 0.60 indicating a negligible effect to a moderate effect. With externalizing outcomes, observational effects ranged from *g* = −0.53 to −0.22 indicating a moderate to small effect. Teacher-rated outcomes ranged *g* = −0.03 to −0.56, indicating a small negligible effect to a moderate effect.

Study design. For teacher outcomes, seven studies were coded as an RCT and one was coded as a QED. For RCT studies, effect sizes ranged from *g* = 0.14 to 2.28 indicating a small to very large effect. Even when removing the outlier, *g* = 2.28 [34], there was still a wide range of effect sizes (*g* = 0.14 to 0.72). The effect size for the QED study was *g* = 0.67 [22], indicating a moderate effect. For negative teacher strategies, effect sizes ranged from *g* = −0.25 to −1.0 for the RCT studies, indicating a small to large effect, while the QED study was *g* = 0.68, indicating a moderate effect [22].

For child outcomes, five studies were coded as an RCT and eight studies were coded as QED. RCT studies ranged between *g* = 0.14 and 0.93 (*N* = 5), indicating a small to large effect on child prosocial skills. However, the effect size of *g* = 0.93 [22] seems to be an outlier as most of the studies range from *g* = 0.14 to 0.30 (*N* = 3). For RCT studies, the values ranged from *g* = −0.24 to 0.60 (*N* = 8), indicating a small to moderate effect; however, most of the studies (*N* = 5) range within the range of *g* = 0.11 to 0.24, indicating a small effect. (See Table 3).

For externalizing outcomes, QED studies ranged between *g* = −0.03 and −0.56 (*N* = 5) indicating a negligible to moderate effect size. However, the effect size of *g* = −0.56 [28] seems to be an outlier as most of the studies were between *g* = −0.03 and −0.15 (*N* = 4). RCT studies ranged from *g* = −0.15 to −0.52, indicating a small to moderate effect size, however, the effect size of *g* = −0.52 [29] seems to be an outlier as most of the studies were between *g* = −0.11 and −0.24.

Dosage. Seven studies included dosage in teacher outcomes. However, six of the seven studies were coded as low dose (See Table 1). For positive teacher strategies, low dosage effect sizes ranged from *g* = −0.15 for to 0.72 indicating a small negative effect to moderate positive effect; however, the high dosage effect size was large (*g* = 2.23). For negative teacher strategies, low dosage ranged from small effect size of *g* = −0.30 to large effect size of −1.0 (*N* = 6). There were no negative teacher strategies reported for this measure, so a comparison could not be made.

Thirteen studies included dosage for child outcomes. Effect sizes for prosocial child outcomes ranged from *g* = 0.16 to 0.60 for high dosage child outcomes (*N* = 4), indicating a small to moderate effect. Low dosage ranged from *g* = −0.24 to 0.92, indicating a small to large effect. For low dosage, *g* = 0.92 [22] seems to be an outlier as four out of the six of these studies had effect sizes between *g =* 0.12 to 0.30. For externalizing outcomes, high dosage ranged from *g* = −0.03 to −0.56 based on four studies. The effect size of *g* = −0.03 [26], seems to be an outlier as 3 out of the 4 studies were between −0.23 and −0.56, indicating a small to moderate effect size. Low dosage ranged between *g* = −0.07 and −0.30 (*N* = 7), indicating a negligible to small effect size. The effect size of *g* = −0.07 [30] seems to be an outlier as most of the studies ranged between *g* = −0.15 and −0.30.

## 4. Discussion

This study was conducted to quantitatively summarize the current state of evidence of IYTCM to improve teacher and child outcomes. Based on the literature search, only three studies examined perceived teacher usefulness of positive classroom management strategies [24,25,38]. Thus, these results were synthesized narratively. Although there was a wide range of effect sizes in these studies (*g* = 0.10 to 0.94), the preliminary results indicated positive outcomes when implementing positive classroom management strategies. However, these effect sizes should be interpreted with caution as there are not enough studies to indicate teacher’s perceived usefulness of positive classroom management strategies. The reasons why studies did not include a measure of perceived usefulness of classroom management strategies is unclear, especially since teachers’ lack of buy-in can be a barrier to successful treatment implementation [40].

Pertaining to the first research question on the impact on teacher outcomes, IYTCM had moderate effects on teachers’ use of positive and negative classroom strategies, indicating IYTCM can improve TCM strategies. Parent strategies were not examined in the previous meta-analysis analyzing the IYPT program [17], but the current findings (*g* = 0.70 to 0.73) demonstrate slightly larger effects than previous meta-analytic reviews examining parenting training (*d* = 0.45; 13; *d* = 0.44; [12]). The reason for the larger positive effect size of positive teacher strategies may be due to the inclusion of [34] in our sample (*g* = 2.27). This study was conducted in an urban area in Jamaica where most teachers only had a high school degree and little training in TCM. Substantial improvements may have been because these teachers were starting out with less experience, and thus had more room for improvement. Even without this study, the current study findings pertaining to teacher outcomes parallel previous research on the IYPT. The implications of these findings strengthen the existing empirical research that training teachers can improve positive TCM strategies and reduce negative TCM strategies.

Pertaining to the second research question on the impact of IYTCM on child outcomes, on average, small positive effects were found on children’s externalizing (*g* = −0.14 to −0.16) and prosocial behavior (*g* = 0.19 to 0.21) for teacher-rated reports. These findings are similar to previous meta-analytic reviews on the IYPT program (disruptive behavior: *d* = −0.27; prosocial behavior: *d* = 0.23; 17). The results of this study strengthen the existing empirical research that training caregivers (i.e., teachers, parents) is essential to improving both teacher [10,42] and child outcomes [13]. Past research supports the use of parent training programs to improve child behavior but has included less research to support teacher training as an effective intervention. The positive findings of this study show that training teachers to use effective TCM strategies can also improve child prosocial and externalizing behavior such as the parent version.

We examined target child, dosage, reporting method, and study design as potential moderators to investigate the high levels of heterogeneity in both teacher and child outcomes, however, due to a limited number of studies and missing data, empirical analysis was not possible. Descriptively speaking, more studies included larger effect sizes for high-risk children and higher dosage (≥42 h), which aligns with findings from previous meta-analytic reviews [17] and previous IYTCM studies comparing clinical severity of child behavior [22,25]. In contrast, both observation (prosocial: *g* = 0.13 to 0.93; externalizing: *g* = −0.53 to −0.22) and teacher-rated effect sizes (prosocial: *g* = −0.03 to 0.60; externalizing: *g* = −0.03 to −0.56) varied from small negative effects to large positive effects, contradicting the meta-analysis on the IYPT program which found larger effects in observation compared to teacher-rated reports. Finally, for study design, the studies coded as RCT seemed to have larger effect sizes than QEDs, which also contradicted the previous meta-analysis on IYPT [17]. The results from these findings indicate that target child and dosage may influence the effectiveness of IYTCM and indicate a need for further exploration into these variables.

## 5. Limitations

This meta-analytic study contained several limitations. First, we only included published studies and conference abstracts, and excluded dissertations. To address this problem, an assessment for publication bias was intended. However, due to an inadequate number of studies (<10 studies per outcome) this assessment could not be conducted via a funnel plot [45]. Therefore, this study may have a file-drawer problem which can cause an upward biased effect [39]. Second, there were a small number of studies in our sample, which can lower statistical power and make it more difficult to detect a statistically significant effect. Conducting more research on both teacher and child outcomes would provide a meta-analysis with greater power in the future. Finally, results indicated a high percentage of heterogeneity, which may indicate these studies are not from the same population. To explain heterogeneity, our goal was to conduct a moderator analysis, however, due to the small sample, we were unable to analyze which moderators may be explaining these high levels of heterogeneity. Thus, the results of this meta-analysis should be interpreted with caution.

## 6. Implications for Research and Practice

IYTCM is designed to influence children’s behavior through the improvement of TCM strategies. Since caregivers have a considerable effect on child development, IYTCM is especially relevant in preventing and improving behavioral issues. Despite the limitations, this meta-analysis summarizes the current state of literature and demonstrates IYTCM has positive effects on teacher and child outcomes. The results of this meta-analysis indicate IYTCM produces similar caregiver effects when compared to other parent behavioral training meta-analyses, and similar results for child outcomes when compared to the IYPT program. These results strengthen the argument that caregiver training can promote positive influences on child development across settings. The positive outcomes of IYTCM on child behavior also have important implications for the schools setting. As classroom-based problems may not be readily managed through parent skill training and parents may not be accessible to address challenging classroom behaviors, IYTCM could help bridge this gap. Furthermore, positive effects on TCM strategies indicates a potential benefit for teachers lacking classroom management training. With improved classroom management skills, IYTCM may prevent teacher burnout and improve teacher retention rates [2].

Results regarding the effect size and the number of RCTs included in this research provide support for IYTCM as an EBI. However, the parts of this program that work best and for whom, follow-up effects, and acceptability of the program remain unclear. Specifically, future researchers should compare high-risk children to low-risk children as only four studies were identified in this study. This could help determine if IYTCM can be used as a universal intervention or if it works best for high risk populations. Additionally, few studies contained follow up information. Thus, future researchers should attempt to examine this outcome to see if IYTCM has long lasting effects on both teacher and child outcomes. Finally, because we found less studies examining teacher outcomes and only three studies examining teacher perceived usefulness of the TCM studies, we recommend researchers examine these variables in the future.

## Figures and Tables

**Figure 1 children-09-00024-f001:**
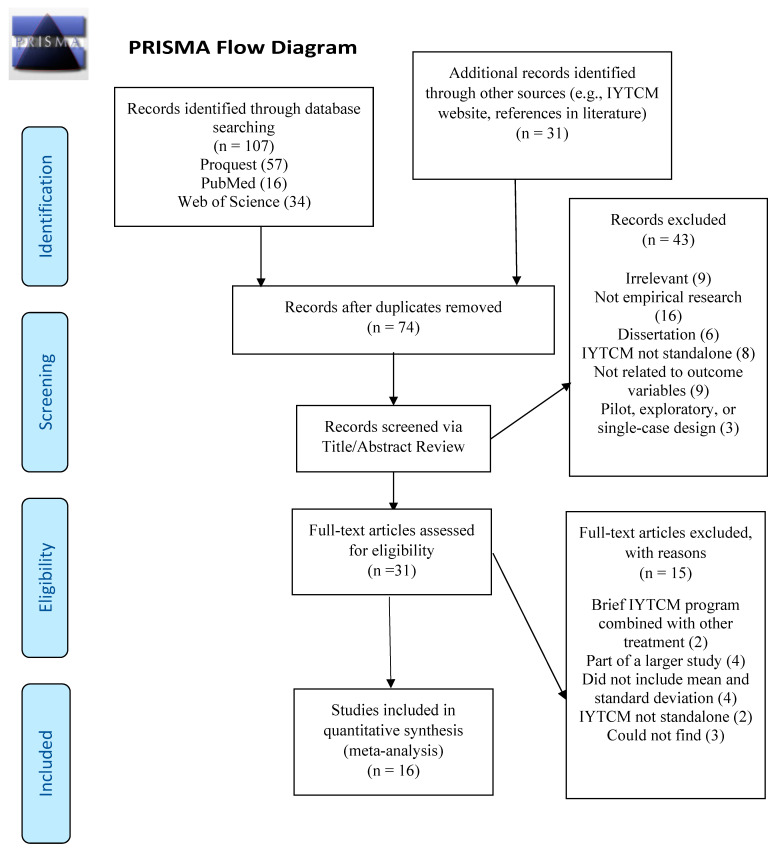
Preferred Reporting Items for Systematic Reviews and Meta-Analyses (PRISMA) Flow Diagram. IYTCM = Incredible Years Teacher Classroom Management.

**Table 1 children-09-00024-t001:** Summary Characteristics of Teacher and Child Outcomes.

Study	Outcome Variable	Study Design	Dosage	Target Child	Reporting Method	Quality of Study
Aasheim et al. (2018) [26]	Positive Teacher Outcomes Negative Teacher Outcomes	QED	High Dosage	Low Risk	TR	7
Baker-Henningham et al. (2012) [29]	Positive Teacher Outcomes Negative Teacher Outcomes	RCT	High Dosage	High Risk	Obs. TR	8
Baker-Henningham & Walker (2018) [33]	Positive Teacher Outcomes	RCT	High Dosage	NA	NA	7
Carlson et al. (2011) [38]	Perceived Usefulness	QED	Low Dosage	NA	NA	2
Fergusson et al. (2013) [24]	Perceived Usefulness	QED	High Dosage	NA	NA	2
Ford et al. (2018) [14]	Prosocial Behavior Externalizing Behavior	RCT	High Dosage	Low Risk	TR	4
Fossum et al. (2017) [27]	Prosocial Behavior Externalizing Behavior	QED	Low Dosage	Low Risk	TR	5
Hickey et al. (2017) [25]	Positive Teacher Outcomes Negative Teacher Outcomes Perceived Usefulness Prosocial Behavior Externalizing Behavior	RCT	Low Dosage	Low Risk	Obs. TR	6
Hutchings et al. (2007) [22]	Perceived Usefulness Positive Teacher Outcomes Negative teacher Outcomes	QED	Low Dosage	High Risk	Obs.	2
Hutchings et al. (2013) [23]	Positive Teacher Outcomes Negative Teacher Outcomes	RCT	Low Dosage	Low Risk	Obs.	8
Kirkhaug et al. (2016) [28]	Prosocial Behavior Externalizing Behavior	QED	High Dosage	High Risk	TR	4
Murray et al. (2017) [36]	Positive Teacher Outcomes Prosocial Child Outcomes	RCT	Low Dosage	Could not tell	TR	7
Murray, Murr, & Rabiner (2012) [34]	Positive Teacher Outcomes Negative Teacher Outcomes	RCT	Low Dosage	Could not tell	TR	6
Murray, Rabiner, & Carrig (2012) [35]	Positive Teacher Outcomes	RCT	Low Dosage	NA	NA	4
Raver et al. (2008) [31]	Positive Teacher Outcomes Negative Teacher Outcomes	RCT	Low Dosage	NA	NA	7
Seabra-Santos et al. (2018) [30]	Prosocial Behavior Externalizing Behavior	QED	Low Dosage	Low Risk	TR	4

Note. RCT = randomized controlled trial; QED = quasi-experimental design; high dosage = received 42 h or more of IYTCM training; low dosage = less than 42 h of IYTCM training; Obs. = observation; TR = Teacher-rating form; NA = Not applicable.

**Table 2 children-09-00024-t002:** Effect size results of Teacher and Child Outcomes.

	ρ = 0	ρ = 0.2	ρ = 0.5	ρ = 0.7	ρ = 0.95
Outcome Variable	*g* (*SE*) 95% CI [LL, UL]	*g* (*SE*) 95% CI [LL, UL]	*g* (*SE*) 95% CI [LL, UL]	*g* (*SE*) 95% CI [LL, UL]	*g* (*SE*) 95% CI [LL, UL]
Teacher Positive	0.70 (0.27) ** [0.17, 1.24]	0.71 (0.27) ** [0.18, 1.24]	0.72 (0.27) ** [0.19, 1.24]	0.72 (0.26) ** [0.20, 1.24]	0.73 (0.26) ** [0.22, 1.24]
Teacher Negative	−0.59 (0.16) *** [−0.90, −0.27]	−0.57 (0.15) *** [−0.86, −0.28]	−0.53 (0.13) *** [−0.79, −0.28]	−0.52 (0.12) *** [−0.76, −0.28]	−0.50 (0.11) *** [−0.72, −0.27]
Child Prosocial	0.19 (0.06) ** [0.07, 0.32]	0.20 (0.06) ** [0.07, 0.32]	0.20 (0.06) ** [0.08, 0.33]	0.20 (0.06) ** [0.08, 0.33]	0.21 (0.06) ** [0.08, 0.33]
Child Externalizing	−0.14 (0.07) * [−0.28, −0.01]	−0.14 (0.07) * [−0.29, −0.01]	−0.15 (0.07) * [−0.29, −0.01]	−0.15 (0.07) * [−0.30, −0.01]	−0. 16 (0.06) * [−0.30, −0.01]

*** *p* < 0.001 ** *p* < 0.01, * *p* < 0.05; Confidence Interval (CI): 95% confidence interval. LL = lower limit; UL = upper limit.

**Table 3 children-09-00024-t003:** Effect Sizes for IYTCM on Teacher and Child Outcomes.

	Child Prosocial		Child Externalizing		Teacher Positive	Teacher Negative	Teacher Usefulness
	Obs.	TR	Obs.	TR			
Article	*g* (*SE*)	*g* (*SE*)	*g* (*SE*)	*g* (*SE*)	*g* (*SE*)	*g* (*SE*)	*g* (*SE*)
Aasheim et al. (2018) [26]		0.16 (0.006)		−0.03 (0.032)			
Baker-Henningham et al. (2012) [29]	0.29 (0.08)	0.60 (0.08)	−0.23 (0.082)	−0.514 (0.09)			
Baker-Henningham & Walker (2018) [33]					2.3 (0.3)		
Carlson et al. (2011) [38]							0.78 (0.23)
Fergusson et al. (2013) [24]							0.88 (0.08)
Ford et al. (2018) [14]		0.20 (0.045)		−0.15 (0.045)			
Fossum et al. (2017) [27]		0.13 (0.06)		−0.12 (0.045)			
Hickey et al. (2017) [25]	0.14 (0.14)	0.12 (0.13)	−0.26 (0.14)	−0.29 (0.14)	0.48 (0.42)	−10.0 (0.44)	0.94 (0.44)
Hutchings et al. (2007) [22]	0.92 (0.67)		−0.15 (0.65)				
Hutchings et al. (2013) [23]	−0.25 (0.19)		−0.3 (0.19)		0.69 (0.55)	−0.46 (0.54)	
Kirkhaug et al. (2016) [28]		0.29 (0.22)		−0.56 (0.22)			
Murray et al. (2017) [36]		00.03 (0.06)			0.15 (0.15)	−0.26 (0.22)	
Murray, Murr, & Rabiner (2012) [34]					0.44 (0.22)	−0.42 (0.20)	
Murray, Rabiner, & Carrig (2012) [35]		−0.03 (0.06)			0.23 (0.02)	−0.34 (0.20)	
Raver et al. (2008) [31]					0.73 (0.02)	−0.66 (0.21)	
Seabra-Santos et al. (2018) [30]		0.14 (0.05)		−0.07 (0.048)			

Note. Obs. = Observation of child outcomes; TR = Teacher rating form of child outcomes.

## Data Availability

The data presented in this study can be obtained by contacting the first author.

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
