# Peer review of "A Meta-Analysis of the Current State of Evidence of the Incredible Years Teacher-Classroom Management Program"

_children, 2021, doi:10.3390/children9010024_

Round 1

Reviewer 1 Report

Dear authors,

Congratulations for your work. You master this field and present an interesting paper. 

In the Theoretical Background you state two aims and three research questions. In my opinion, this could fit better in a more extensive introduction or in the method section. In addition, you respond to this statement in the discussion, but I would suggest to add a section of Conclusions to give answer more directly and to explicit what's your contribution with this work (it could ease to be cited).

It appears justified, but the discussion section contrasts the results with just four different references. If possible, contrasting with further literature would make your discussion of results even more consistent.

Revise the references list according to MDPI guidelines. For instance, journal names must be shortened. Link: https://www.mdpi.com/authors/references 

And citations must be numerical, not parentetical with authors' names: https://mdpi-res.com/data/mdpi_references_guide_v4.pdf 

Author Response

Reviewer #1:

Thank you so much for your positive feedback about our manuscript. I note that we have conducted a spell check per your recommendation in addition to addressing each of your concerns raised below. We have provided a point-by-point overview of the changes made (appearing in track changes). We believe these changes to the Introduction and Conclusions have substantially improved our manuscript and thank you for those suggestions.

  1. In the Theoretical Background you state two aims and three research questions. In my opinion, this could fit better in a more extensive introduction or in the method section. In addition, you respond to this statement in the discussion, but I would suggest to add a section of Conclusions to give answer more directly and to explicit what's your contribution with this work (it could ease to be cited).

Response: We have combined the Introduction and Theoretical Background sections based on your feedback. The Introduction is now more extensive per your recommendation. In addition, this change resulted in the aims/research questions appearing in their own section (no longer subsumed under Theoretical Background) which we agree helps to set the stage for the Methods section. We have also made the number of aims consistent with our research questions to prevent any confusion.

  1. It appears justified, but the discussion section contrasts the results with just four different references. If possible, contrasting with further literature would make your discussion of results even more consistent.

Response: Thank you for pointing out the contrast between the references we cited in the results section compared to the discussion. We note that the results section was so heavily focused on the references of associated with the articles used in our meta-analyses. Thus, we could not reduce those. Instead, we have edited the manuscript to more fully discuss our meta-analytic results in the context of the prior meta-analytic studies completed on the IYPT version while also calling attention to a few of the IYTCM articles. We hope you find that our reference additions to the discussion have now addressed your feedback.

  1. Revise the references list according to MDPI guidelines. For instance, journal names must be shortened. Link: https://www.mdpi.com/authors/references 

Response: We have made those changes to the reference list consistent with MDPI guidelines.

  1. And citations must be numerical, not parentetical with authors' names: https://mdpi-res.com/data/mdpi_references_guide_v4.pdf 

Response: We have numbered our citations consistent with the MDPI guidelines.

Reviewer 2 Report

It is a well-organized and structured work. Interesting in the subject it deals with and enjoyable in its presentation.

Proposed amendments:

INTRODUCTION. Indicate meaning of acronyms (IY: Incredible Years).

The section VARIABILITY WITHIN THE RESEARCH, can be included in section METHOD. It is justified why a meta-analysis to achieve the proposed objectives.

Two objectives are proposed: To examine the current state of evidence regarding teacher and child outcomes. Consider adding: "in the use of IYTCM techniques".

There were two aims of the current study. First, we examined the current state of evidence with respect to teacher and child outcomes “in the use of IYTCM techniques”.

Three research questions are specified and three working hypotheses are included.

Based on previous meta-analytic research and systematic reviews on parent training [line 165] programs (Menting et al., 2013; Lundahl, Risser, & Lovejoy, 2006), we hypothesized that (a) IYTCM would have small to moderate effects on teacher outcomes overall, (b) IYTCM would have small to moderate effects on child outcomes overall, and c) the target child, reporting methods, study design, and dosage would not moderate the overall effect size of IYTCM

Refer in section DISCUSSION to the working hypotheses.

When the objectives are specified, it is also specified what will be explored for children and teachers. (line 145-147). It may be more appropriate to include this information in section METHOD.

Line 201-203, it says: This resulted in a total of 16 articles. For a visual detail overview of the articles located/ excluded, see Figure 1.

Replace with: This resulted in a total of 16 articles (Figure 1).

Title Figure 1. Adapted from Moher et al. (2006). The reference is already in its section.

-Coding of the study. Identify in section Study design (line 255-259) the meaning of acronyms RCT and QED.

As a study, CHANGE Lundahl and colleagues (2006) BY Lundahl et al. (2006).

META-ANALYSIS CALCULATIONS. Indicate that the R MVMETA package USES MULTIVARIATE REGRESSION TECHNIQUES AS A SYSTEM OF ANALYSIS.

Author Response

Reviewer #2:

Thank you so much for the positive feedback pertaining to our manuscript. We appreciated your feedback on the organization and structure. We also were happy to read that you thought the work was interesting and enjoyable to read. Please see below for the point-by-point edits made consistent with your feedback.

  1. Indicate meaning of acronyms (IY: Incredible Years).

Response: We have added the meaning of this acronym in the Abstract and Introduction (first time acronym is used)

  1. The section VARIABILITY WITHIN THE RESEARCH, can be included in section METHOD. It is justified why a meta-analysis to achieve the proposed objectives.

Response: After much reflection we have decided that the Variability within the Research section in the Introduction helps to set up Aim/Research Question #3. Thus, we have decided to keep it in that section, though we did delete the first sentence as it did not fit well in that section and we altered the wording of the last sentence to improve readability.

  1. Two objectives are proposed: To examine the current state of evidence regarding teacher and child outcomes. Consider adding: "in the use of IYTCM techniques".

Response: We have altered the wording of our three aims to reflect this suggestion to link the IYTCM techniques with the outcomes expected.

  1. There were two aims of the current study. First, we examined the current state of evidence with respect to teacher and child outcomes “in the use of IYTCM techniques”.

Response: We have altered the wording of our three aims to reflect this suggestion to link the IYTCM techniques with the outcomes expected.

  1. Three research questions are specified and three working hypotheses are included.

Response: We have now linked those to our three aims more directly. In addition, we explicitly refer to these within the discussion section.

  1. Based on previous meta-analytic research and systematic reviews on parent training [line 165] programs (Menting et al., 2013; Lundahl, Risser, & Lovejoy, 2006), we hypothesized that (a) IYTCM would have small to moderate effects on teacher outcomes overall, (b) IYTCM would have small to moderate effects on child outcomes overall, and c) the target child, reporting methods, study design, and dosage would not moderate the overall effect size of IYTCM

Response: We were unclear what edit was requested in feedback #6. Thanks for that additional guidance as we are happy to make it.

  1. Refer in section DISCUSSION to the working hypotheses.

Response: We have made references in the discussion to the research questions. Thanks for that suggestion as it was consistent with other reviewers’ feedback.

  1. When the objectives are specified, it is also specified what will be explored for children and teachers. (line 145-147). It may be more appropriate to include this information in section METHOD.

Response: After much consideration about this piece of feedback, we have felt it is imperative to discuss what will be explored for children and teachers in the objectives section to allow readers to see the specific outcomes to be examined within our three research questions.

  1. Line 201-203, it says: This resulted in a total of 16 articles. For a visual detail overview of the articles located/ excluded, see Figure 1.

Replace with: This resulted in a total of 16 articles (Figure 1).

Response: Thank you for that suggestion. That edit has been made

  1. Title Figure 1. Adapted from Moher et al. (2006). The reference is already in its section.

Response: Per your feedback, we have deleted the reference from Figure 1.

  1. Coding of the study. Identify in section Study design (line 255-259) the meaning of acronyms RCT and QED.

Response: Randomized controlled trial (RCT) was previously identified on line 126 and quasi-experimental designs (QED) was indicated in line 147.

  1. As a study, CHANGE Lundahl and colleagues (2006) BY Lundahl et al. (2006).

Response: Thank you. This edit has been made.

  1. META-ANALYSIS CALCULATIONS. Indicate that the R MVMETA package USES MULTIVARIATE REGRESSION TECHNIQUES AS A SYSTEM OF ANALYSIS

Response: Thank you for that edit, it has been made in the manuscript.

Reviewer 3 Report

First of all, I would like to congratulate the authors for such a commendable work. The paper shows an adequate and deep scientific process that has a sufficient level to be published. However, I would like to recommend small improvements necessaries to improve the final product of this paper:

  1. Firstly, the theoretical justification on 'classroom management' seems to me insufficient and out of date. It could be updated with various current references like the ones  I present below (among others).
  • Barrientos et al. (2020) Social and Emotional Competences of the Teacher of Childhood Education and its Relationship with Classroom Management: https://doi.org/10.15581/004.38.59-78 
  • Lee and van Vlack (2017) Teachers’ emotional labour, discrete emotions, and classroom management self-efficacy. https://doi.org/10.1080/01443410.2017.1399199
  • Weber et al (2018) Promoting pre-service teachers' professional vision of classroom management during practical school training: Effects of a structured online- and video-based self-reflection and feedback intervention. https://doi.org/10.1016/j.tate.2018.08.008 
  1. The first objective is not understood and would have to be reworked appropriately. In fact, the entire Objectives section needs to be reviewed and adapted to fit the rest of the project carried out, since the research carried out does not correspond 100% to what is described in the objectives, which detracts from the final result of the manuscript.

Nothing more. I repeat my congratulations. Thank you for giving me the opportunity to see your work.

Author Response

Reviewer #3

Thank you so much for your positive feedback on our manuscript. We have improved the Introduction section to provide more sufficient background and relevant references. Thank you so much for those suggestions. Below we summarize point-by-point our response to your feedback. Those changes have substantially improved our work and we thank you for that.

  1. Firstly, the theoretical justification on 'classroom management seems to me insufficient and out of date. It could be updated with various current references like the ones  I present below (among others).
  • Barrientos et al. (2020) Social and Emotional Competences of the Teacher of Childhood Education and its Relationship with Classroom Management: https://doi.org/10.15581/004.38.59-78 
  • Lee and van Vlack (2017) Teachers’ emotional labour, discrete emotions, and classroom management self-efficacy. https://doi.org/10.1080/01443410.2017.1399199
  • Weber et al (2018) Promoting pre-service teachers' professional vision of classroom management during practical school training: Effects of a structured online- and video-based self-reflection and feedback intervention. https://doi.org/10.1016/j.tate.2018.08.008

Response: Thank you for these citations and suggestions. We have altered the theoretical section to reflect a more current literature base. We have also deleted an older reference to Bowlby’s attachment theory given that Patterson’s theory is most relevant to the IY program. We note that all but the Barrientos et al (2020) articles are now cited as I only had access to the Spanish version of that article. Alternatively, we have now cited an article by Sanchez-Cabrero et al. 2021 from Educ. Sci. given its relevance and recency.

  1. The first objective is not understood and would have to be reworked appropriately. In fact, the entire Objectives section needs to be reviewed and adapted to fit the rest of the project carried out, since the research carried out does not correspond 100% to what is described in the objectives, which detracts from the final result of the manuscript.

Response: Thank you for this feedback. Based on other editorial feedback, we have also made the number of aims (objectives) consistent with our research questions to prevent any confusion. Further, clarification was brought to each research question. We have also improved the discussion by clearly highlighting each research question.

Round 2

Reviewer 1 Report

Dear authors,

Thanks for considering my suggestions, for the applied changes and for your reply to my comments.